# Influence of Gender Role on Resilience and Positive Affect in Female Nursing Students: A Cross-Sectional Study

**DOI:** 10.3390/healthcare13030336

**Published:** 2025-02-06

**Authors:** L. Iván Mayor-Silva, Guillermo Moreno, Alfonso Meneses-Monroy, Patricia Martín-Casas, Marta M. Hernández-Martín, Antonio G. Moreno-Pimentel, Leyre Rodríguez-Leal

**Affiliations:** 1Facultad de Enfermería, Fisioterapia y Podología, Universidad Complutense de Madrid, 28040 Madrid, Spain; limayors@ucm.es (L.I.M.-S.); ameneses@ucm.es (A.M.-M.); pmcasas@ucm.es (P.M.-C.); martamah@ucm.es (M.M.H.-M.); antomo05@ucm.es (A.G.M.-P.); 2Grupo de Investigación Cardiovascular Multidisciplinar Traslacional (GICMT), Instituto de Investigación Hospital 12 de Octubre (imas12), 28041 Madrid, Spain; 3Health Research Institute of the San Carlos Clinical Hospital of Madrid (IdISSC), 28040 Madrid, Spain; 4Red Cross Nursing University College, Autonomous University of Madrid, 28003 Madrid, Spain; leyre.rodriguez@cruzroja.es

**Keywords:** gender role, women, nursing students, psychological resilience, affect

## Abstract

**Introduction**: Women experience more social barriers, gender stereotypes, biases, and discrimination than men, which can increase their vulnerability to mental health problems. Therefore, it is essential to adopt a gender perspective in research on nursing students, examining the impact of these factors on their well-being and psychological resources like resilience. This study aims to analyze the relationship between gender roles in resilience and positive or negative affect among female nursing students. **Methods**: A cross-sectional study was conducted with first- and fourth-year female nursing students at a public university in Madrid, Spain. Sociodemographic variables, positive and negative affect (PANAS scale), resilience (CD-RISC scale), and gender roles (BRSI inventory) were analyzed. ANOVA, correlation analysis, and linear regression models were used to study the relationships between variables. **Results**: The study included 338 students with a mean age of 21.43 years, of which 80.2% had a high level of resilience, with a positive affect score of 31.96 (SD: 7.34) and a negative affect score of 22.99 (SD: 7.35). Overall, 48.5% had undifferentiated roles, 23.7% feminine roles, 14.2% androgynous roles, and 13.6% masculine roles. Female students with masculine and androgynous roles showed higher resilience levels (93.48% and 97.92%) compared to those with feminine and undifferentiated roles (81.25% and 70.73%) (*p* < 0.001). Female students with androgynous and masculine roles showed higher positive affect levels compared to those with feminine and undifferentiated roles (*p* < 0.001), with no differences in negative affect. These results were observed in both first- and fourth-year students. A high correlation was found between masculine roles and positive affect and resilience in both first- and fourth-year students. **Conclusions**: Gender roles influence positive affect and resilience in females. Among female nursing students, androgynous and masculine roles are associated with higher levels of resilience and positive affect compared to feminine and undifferentiated roles. Differences in psychological well-being may be related to socially constructed gender roles rather than biological sex, with masculine roles enhancing resilience and feminine roles correlating with greater vulnerability.

## 1. Introduction

Gender refers to the prototypical characteristics of men and women that are socially constructed. This includes norms, behaviors, and roles [1]. According to the Gender Role Congruence Theory [2,3], society tends to assign certain jobs predominantly to men or women. This is due to the association of specific traits with each gender [4]. Women make up the majority gender in the nursing profession, comprising around 70% of the workforce [5]. The nursing profession has historically been subject to gender biases due to the social perception that women are inherently more suited for caregiving and nurturing roles, which have traditionally been associated with femininity [6]. These stereotypes have reinforced the notion that women, because of their “maternal” and “compassionate” nature, are more suited to perform caregiving tasks, while men are typically associated with more technical or authoritative roles in other areas of healthcare [7]. Such biases have shaped not only the demographic composition of the profession but also the social and economic recognition of nurses, who have often been valued less than other healthcare professionals [8]. Although the professionalization and diversification of nursing have challenged these stereotypes, the legacy of gendered roles continues to influence both the perception and practice of the profession [7].

Resistance to change and the persistence of stigmas can uphold certain prejudices and barriers that affect women who take on the role of being nursing students, causing additional stress and negative affectivity [9]. The development of resilience, understood as the ability to cope with and recover from adverse situations [10], through support programs and mentoring, as well as training in emotional intelligence, could help nursing students better manage emotional challenges and reduce negative affectivity [11].

Affect, encompassing both positive and negative emotions, plays a pivotal role in shaping individuals’ mental and physical well-being. Positive affect, characterized by feelings of joy, enthusiasm, and contentment, is frequently linked to higher levels of psychological well-being and resilience [12]. Conversely, negative affect, which includes emotions such as anxiety, anger, and sadness, has been consistently associated with poorer mental health outcomes, including increased vulnerability to depression and anxiety disorders [13]. The impact of affect on well-being is multifaceted, with both emotional states influencing individuals’ capacity to cope with stress and their overall quality of life [14].

Women tend to experience both positive and negative emotions more intensely than men [15], especially among younger women [16]. They also experience more mental health issues, such as depression or low mood, with a higher proportion of severe symptoms in certain forms of bipolar depression [17]. This heightened emotional responsiveness in women can influence their mental health, as increased negative affect may contribute to greater psychological distress.

Women are more likely to experience socialization processes that reinforce traditional gender roles, such as caregiving and emotional expressiveness, which can exacerbate the negative impact of affective states on mental health [18,19]. Moreover, women face greater imposition of social barriers, gender stereotypes, biases, and discrimination than men [20], which may also contribute to a greater vulnerability among women, starting at a young age, to mental health problems. In contrast, women who adopt more masculine or androgynous roles, which often emphasize assertiveness and emotional control, may report higher levels of resilience and psychological well-being, possibly due to the social empowerment associated with these roles [21].

Negative affect has been associated with increased stress, which can manifest as physical symptoms such as fatigue, muscle tension, and cardiovascular problems [22]. Women, particularly those who internalize traditional gender norms, may experience greater physical health issues due to the compounded effects of negative emotions and gendered stressors [23]. Positive affect, on the other hand, has been linked to improved immune function, better cardiovascular health, and lower levels of inflammation, with these benefits often mediated by the adoption of less restrictive gender roles [24].

Studies on gender roles in nursing students have focused on analyzing the impact of valuing the gender perspective on the development and evolution of the profession [25] or on studying changes over time in stereotypes about nursing among students [26]. However, few have examined the impact of gender roles on students’ emotional well-being, and even fewer have explored their connection to resilience. This highlights the need for research that incorporates a gender lens, particularly in relation to how gender-related social factors affect women’s well-being. Additionally, it is important to investigate how these factors influence psychological resources, such as resilience, which can help protect women from mental health challenges in nursing practice [27].

This study aims to analyze the influence of gender roles on the level of resilience and on the levels of positive and negative affectivity in female nursing students, as well as to explore sociodemographic factors, such as academic year, that could explain these potential differences.

## 2. Material and Methods

### 2.1. Design

A cross-sectional study was conducted with female nursing students from the first and fourth years of a public university in Madrid (Spain), using the STROBE (STrengthening the Reporting of OBservational studies in Epidemiology) statement for cross-sectional studies (2008) as a framework. We selected this university population to understand how young adult women, at a key stage of personal and professional development, perceive gender roles and how these perceptions impact their emotional well-being and resilience. Selecting first- and fourth-year students provided an opportunity to explore how the influence of gender roles evolves with personal and professional development. Previous research has indicated differences in resilience and positive affect between first- and fourth-year nursing students [28]. Thus, investigating how gender roles impact students at different stages of growth can help determine whether the effects of social stereotypes are independent of the maturation process.

### 2.2. Participants

The recruitment of students took place during theoretical classes for both first-year and fourth-year students, where the study’s objectives were explained.

### 2.3. Study Variables

The study variables included sociodemographic variables (sex, marital status, date of birth, current living situation, paid employment or not, volunteer activities, occasional work, high-level or federated sports), the participants’ academic year (first or fourth), and the following main variables:-Positive and negative affect: Measured using the Positive and Negative Affect Schedule (PANAS) [13] in its Spanish version [24]. The PANAS questionnaire is a self-report instrument that assesses an individual’s affective state [29]. It consists of two subscales, each with 10 items, which are answered using a 5-point Likert scale, where 1 corresponds to “not at all” and 5 corresponds to “extremely”. One scale measures the positive affect or recent positive experiences, which act as a protective factor, while the other measures negative affect or experiences, which may act as a risk factor for diseases. The scales have demonstrated consistency and stability in the Spanish university population, with good construct validity and a high reliability index (Cronbach’s alpha: >0.87) [30]. In our sample, it maintains acceptable reliability with a Cronbach’s alpha of 0.755.-Resilience: Measured using the Connor–Davidson Resilience Scale (CD-RISC) [31] in its Spanish version [32], which assesses resilience with 25 items using a Likert-type response format with 5 response options (0 = “not at all”, 1 = “rarely”, 2 = “sometimes”, 3 = “often”, and 4 = “almost always”). The scale range is from 0 to 100, with higher scores indicating greater levels of resilience. The thresholds are as follows: below 70 (low); 70 to 87 (medium); above 88 (high). The internal consistency of the Spanish version is optimal, with a Cronbach’s alpha of 0.86 [33] and 0.832 calculated for our sample. The items are grouped into five dimensions: persistence, tenacity, and self-efficacy; control under pressure; adaptability and networks; control and purpose; and spirituality.-Gender roles: Measured using the Bem Sex-Role Inventory (BSRI) [34], an instrument designed to assess an individual’s adherence to stereotypical characteristics of feminine and masculine gender roles. The 60 items of the instrument are scored using a seven-point Likert scale. It consists of 60 adjectives, of which 20 are stereotypical masculine roles, 20 are feminine roles, and 20 are gender-neutral roles. Respondents are asked to indicate the extent to which each item describes them, on a scale from 1 (never) to 7 (always). The instrument classifies individuals as masculine, feminine, androgynous (both masculine and feminine roles), or undifferentiated roles (neither predominantly masculine nor predominantly feminine roles). Once completed, the scores for items 1, 4, 7, 10, 13, 16, 19, 22, 25, 28, 31, 34, 37, 40, 43, 46, 49, 55, and 58 are summed and divided by 20 to obtain the score for masculine roles. Similarly, the scores for items 2, 5, 8, 11, 14, 17, 20, 23, 26, 29, 32, 35, 38, 41, 44, 47, 50, 53, 56, and 59 are summed and divided by 20 to obtain the feminine roles score. The internal consistency of the scale is adequate, with Cronbach’s alpha coefficients for each dimension ranging from 0.75 to 0.90 [35], and a global reliability of 0.78 for the Spanish version (which is the one used in this study) [36] in our case increasing up to 0.844 which generates good reliability.

### 2.4. Data Collection

The students were contacted virtually through an announcement posted on the virtual campus of a course in which they were enrolled (Moodle platform), where the researchers had no teaching responsibilities. Data collection was conducted via a Google Forms questionnaire, which was uploaded to the Moodle platform. Students had one month to complete the form, from 1 October to 1 November 2022.

### 2.5. Data Analysis

The personal information of the participants was anonymized using numerical codes to ensure confidentiality. For qualitative variables, absolute and relative frequencies were calculated, and for continuous variables, measures of central tendency and dispersion were computed. The main variables (positive affect, negative affect, and resilience) were analyzed according to sociodemographic characteristics and the academic year of the students, using analysis of variance (ANOVA) and Student’s *t*-test for independent samples, after confirming normality with the Kolmogorov–Smirnov test and verification of the assumption of homogeneity of variance using Levene’s test. A statistically significant relationship between variables was considered when the *p*-value was <0.05. Finally, a correlation analysis and a simple linear regression analysis were conducted to assess the relationship between positive affect, negative affect, and resilience (dependent variables) based on gender roles, sociodemographic variables, and academic year (independent variables). Data analysis was performed using the SPSS 26v^®^ statistical software.

### 2.6. Ethical Considerations

This study was conducted in accordance with the ethical principles for medical research as outlined in the Declaration of Helsinki. Data handling was confidential and complied with Regulation (EU) 2016/679 of the European Parliament and Council of 27 April 2016 on Data Protection (GDPR), as well as Spanish legislation, specifically Organic Law 3/2018 on Personal Data Protection and the Guarantee of Digital Rights. Additionally, the project was approved by the Research Ethics Committee (CEIC) of the Faculty of Nursing, Physiotherapy, and Podiatry at the Complutense University of Madrid.

## 3. Results

### 3.1. Sample Description

A total of 338 students (172 first-year students and 166 fourth-year students) participated in the study, with 37 (3.43%) surveys excluded due to incompleteness. Table 1 shows the sociodemographic characteristics of the sample. The age range was 18–56 years, with a mean age of 21.43 years (SD: 5.37). Most participants were single (81.1%; n = 274), did not engage in any professional work (65.1%; n = 220), and did not participate in activities outside of the university (45.6%; n = 145). Furthermore, 80.2% (n = 271) of the students had a high level of resilience according to the CD-RISC scale, and 48.5% (n = 164) identified as predominantly undifferentiated in terms of gender (Table 1).

Regarding the main variables, in the dimensions of resilience, mean scores were observed as follows: persistence, tenacity, and self-efficacy 3.84 (SD: 0.54); control under pressure 3.47 (SD: 0.45); adaptability and networks 5.12 (SD: 0.56); control and purpose 3.93 (SD: 0.54); and spirituality 3.18 (SD: 1.03). The mean score for positive affect was 31.94 out of 50 (SD: 7.34) and for negative affect was 22.99 out of 50 (SD: 7.35). No statistically significant differences were found in the sociodemographic characteristics between the gender roles defined by the Bem scale (Table 2).

### 3.2. Differences in Negative Affect, Positive Affect, and Resilience According to Gender Roles

Nursing students exhibited higher levels of resilience when they presented predominantly masculine and androgynous roles (93.48% and 97.92%, respectively) compared to those with feminine or undifferentiated roles (81.25% and 70.73%, respectively), with these differences being statistically significant (*p* < 0.001). Additionally, it was observed that nursing students classified as predominantly androgynous or masculine exhibited higher levels of positive affect compared to those with undifferentiated or feminine roles (*p* < 0.001). Conversely, masculine and androgynous roles in students were associated with lower levels of negative affect compared to undifferentiated and feminine roles, although these differences were not statistically significant (*p* = 0.10) (Figure 1 and Table A1).

Regarding the dimensions of resilience, nursing students with androgynous and masculine roles exhibited higher mean levels of persistence, tenacity, self-efficacy (*p* < 0.001), control under pressure (*p* < 0.001), adaptability and networks (*p* < 0.001), control and purpose (*p* = 0.02), and spirituality (*p* = 0.01) compared to those with feminine or undifferentiated roles. The undifferentiated role showed the lowest levels across all subdimensions of resilience (Figure 2 and Table A1).

### 3.3. Differences in Negative Affect, Positive Affect, and Resilience According to Gender Roles Stratified by Academic Year

Students with androgynous and masculine roles have higher scores in resilience (*p* < 0.001) and positive affect (*p* < 0.001) compared to students with feminine and undifferentiated roles, in both academic years. First-year nursing students with feminine and androgynous roles exhibit higher levels of negative affect compared to those with masculine and undifferentiated roles (*p* = 0.01), whereas in the fourth year, no significant differences were observed between groups (*p* = 0.97) (Figure 3 and Table 3).

Students with androgynous and masculine roles exhibited higher levels of persistence, tenacity, and self-efficacy (*p* < 0.001), as well as control under pressure (*p* < 0.001) and adaptability and networks (*p* = 0.02), compared to students with feminine and undifferentiated roles, both in first and fourth years. In the first year, students with androgynous roles showed higher levels of spirituality, followed by students with feminine roles, masculine roles, and finally, undifferentiated roles (*p* = 0.04). These differences were not observed in fourth-year students (*p* = 0.11). Finally, no differences were observed in the control and purpose dimension, both in first-year (*p* = 0.15) and in fourth-year (*p* = 0.13) students (Table 3).

### 3.4. Correlation Analysis

In the correlation analysis between different variables for first-year and fourth-year students, as shown in Table 4, it was found that in the first year, there was a significant positive correlation between masculine roles and positive affect (r = 0.448; *p* < 0.01) and resilience (r = 0.557; *p* < 0.01). These associations were maintained in the fourth year, both in positive affect (r = 0.328; *p* < 0.01) and in resilience (r = 0.461; *p* < 0.01). A weak but significant negative correlation was also found between masculine roles and negative affect in first-year students (r = −0.077; *p* < 0.01), which disappeared in fourth-year students.

Regarding feminine roles, in the first year, a significant positive correlation was observed between feminine roles and negative affect (r = 0.292; *p* < 0.01), which disappeared in the fourth year. A positive association between feminine roles and resilience (r = 0.216; *p* < 0.01) was found in the first year, which was also true for those who were in their fourth year (r = 0.310; *p* < 0.01). Additionally, no association between positive affect and feminine roles was found in the first year, but such an association was observed in the fourth year (r = 0.278; *p* < 0.01).

### 3.5. Predictive Analysis

The multiple linear regression model constructed to predict whether any of the resilience or affect variables were more strongly related to one gender than another did not yield any significant results for resilience (β = −0.005; *p* = 0.947), persistence, tenacity, and self-efficacy (β = −0.229; *p* = 0.670), control under pressure (β = −0.133; *p* = 0.811), adaptability and networks (β = −0.071; *p* = 0.843), control and purpose (β = −0.059; *p* = 0.838), spirituality (β = −0.043; *p* = 0.798), positive affect (β = −0.008; *p* = 0.410), and negative affect (β = −0.017; *p* = 0.056). No associations were found with other sociodemographic factors, including course year.

## 4. Discussion

This study analyzed the relationships of gender roles on resilience and affect in female nursing students. These students generally exhibit high resilience and adopt either undifferentiated or feminine roles. Resilience, as well as positive and negative affect, varies according to gender role. Students with feminine or undifferentiated roles exhibited lower levels of resilience and positive affect. Additionally, these students scored lower on various resilience subdimensions, including persistence, tenacity, self-efficacy, control under pressure, adaptability, self-regulation, purpose, and spirituality, compared to those with androgynous or masculine roles. No differences were found in negative affect.

Psychological characteristics were largely unaffected by course year, except for negative affect and spirituality. First-year students with feminine or androgynous roles showed higher levels of negative affect and spirituality. However, these differences across gender roles disappeared in the fourth-year students.

Our results provide sufficient evidence to assert that gender roles are related to the experience of positive affect and the development of an optimal level of resilience in young women. These findings are consistent with research suggesting that gender roles and gender flexibility can significantly affect emotional well-being [37,38]. For example, studies have indicated that individuals with androgynous traits, combining both masculine and feminine characteristics, tend to show greater psychological adjustment and well-being [39,40].

Regarding resilience levels, studies suggest that masculinity in young adults acts as a stress buffer, promoting higher social support and resilience by mitigating the negative effects of life event stress. This helps explain why individuals, including women with higher levels of masculinity, often exhibit greater resilience. In contrast, femininity traits have been linked to greater vulnerability to experimentally induced depression. Traits such as excessive concern for others’ welfare, at the expense of one’s own needs, and the pressures imposed by certain contexts may increase susceptibility to depressive feelings, making individuals with higher femininity traits more prone to emotional distress [41].

Consistent with our findings, some studies suggest that adherence to traditional gender roles plays a significant role in the psychological well-being of women, with masculinity emerging as the most important factor influencing the well-being of both women and men [42]. Other studies have shown that masculine roles, rather than feminine roles, serve as protective factors for mental health in university students [43]. Our results also confirm the results obtained by Wood et al. that women who adopt more masculine or androgynous roles report higher levels of resilience and psychological well-being [21].

Some studies have explored whether specific aspects of gender influence well-being, linking traditional gender role patterns and attitudes—such as the privileging of men’s roles in paid work and as “breadwinners,” along with the expectation that women should prioritize caregiving and family responsibilities over other roles—with well-being outcomes. More traditional gender roles and attitudes, for both men and women, have been associated with higher psychological distress, particularly among unemployed men [44]. In this context, it would be valuable to explore whether this dynamic affects young female university students, who must navigate academic demands in an environment that may contrast with their traditional aspirations. Examining this as a potential explanatory mediational factor between femininity roles and lower positive affect and resilience could offer deeper insights into the psychological well-being of these individuals.

The lower score in positive affect and higher score in negative affect among first-year students with a feminine role may reflect the pressures of gender stereotypes and societal expectations [2]. Previous studies have indicated that women tend to internalize gender roles (femininity) to a greater extent than men, a tendency associated with heightened fear of negative evaluation and increased regret in decision-making, ultimately leading to lower well-being [45].

Our findings reinforce the importance of gender identity as a social factor related to emotional states (masculine roles with positive affect and resilience and feminine roles with negative affect in university students), which is consistent with previous research, as well as with the role that high levels of resilience play in the development of a good level of well-being [31]. Furthermore, some studies have suggested that gender is associated with the burden of illness, reinforcing the notion that the internalization of gender roles may significantly influence not only emotional states but also the risk of developing mental disorders. Women exhibit higher rates of lifetime diagnoses for anxiety disorders and are more likely than men to be diagnosed with additional conditions, such as bulimia nervosa and major depressive disorder [46].

The disappearance of the association between negative affect and gender roles in fourth-year students, or the fact that positive affect is only associated with feminine roles in fourth-year students, may be related to the psychological impact that degree training has on nursing students. Other studies have observed that nursing students increase their levels of resilience and positive affect while decreasing negative affect from their first to fourth year of nursing [28]. This suggests that the training students receive may influence how gender roles affect their psychological well-being. This finding is promising, as it implies that the impact of gender-related pressures on psychological well-being and resilience can be mitigated through education and training. Such insights open up significant opportunities for educational interventions aimed at nursing students, helping to promote their mental well-being and enhance their stress-coping skills.

Despite the findings from hypothesis testing statistical methods between groups, we did not find associations between gender roles and the variables of resilience, positive affect, and negative affect in the multiple linear regression models. However, other studies have found that multiple psychosocial factors contribute to emotional well-being and the prediction of students’ mood states [47,48,49]. This could suggest that these traits and emotions are relatively stable across different genders and sociodemographic contexts of this sample. However, further research with larger sample sizes or different analytical approaches might be needed to confirm these findings.

### 4.1. Limitations

Despite the results obtained, this study has several limitations that should be considered. First, it is a cross-sectional study, which prevents establishing cause-and-effect relationships, indicating that caution is needed in the interpretation of the results obtained from our multiple linear regression models. Therefore, our findings should be compared with longitudinal studies conducted with first-year nursing students, followed over the four years of their degree, to more accurately identify changes in the distribution of affect and resilience over time, as well as to precisely determine whether social and gender factors are determinative in this distribution. On the other hand, choosing women as the sample reflects the reality of nursing, but it does not consider the percentage of students (and professionals) who are men, to whom these results cannot be generalized. The genders used in this study are those defined by Bem (male, female, androgynous, and undifferentiated) [37], but there is currently a great diversity of genders beyond those defined by this author and which have been grouped under the terms “genderqueer” or “non-binary” genders [50]. Future studies should consider these other genders in the longitudinal relationship between gender and mental well-being. Finally, the limitation of the sociodemographic data collected, which could be expanded to analyze potential predictors that explain other social factors influencing the unequal distribution of positive and negative affect and resilience capacity. We propose including ethnicity as a key variable in future studies to explore its mediating role in the relationship between gender roles, positive and negative affect, and resilience. Considering cultural differences in emotional expression and coping strategies, ethnicity may offer valuable insights into how these factors interact across diverse populations.

### 4.2. Future Research Directions and Strengths of the Study

This study has the potential to make a significant contribution to the field of nursing by providing empirical data on the relationship between gender roles, affect, and resilience. The findings could have important implications for clinical practice and the training of healthcare professionals, as it suggests that female nursing students with female and undifferentiated roles experience greater gender-related pressure. In addition, our findings may point the way to the development of more effective strategies to support the psychological development and well-being of university students, taking into account the relationship with social role pressures (particularly feminine roles) [51]. The results obtained could be used to improve the academic environment for students by promoting the integration of gender and resilience perspectives in diverse educational and cultural contexts with the aim of creating a more inclusive environment. Additionally, this study could stimulate future research that explores gender dynamics and their impact on mental health within various educational and cultural contexts [52]. Interventions aimed at strengthening resilience and fostering a more flexible gender identity could have beneficial effects on students’ emotional well-being throughout their academic career, as previous research has indicated [37,53].

## 5. Conclusions

Gender roles are related to positive affect and resilience in women. Among nursing students, androgynous and masculine roles are associated with higher levels of resilience and positive affect compared to feminine and undifferentiated roles. Further studies are needed in other populations of female university students and with longitudinal designs to validate our results. It is essential to recognize that the variation in psychological well-being observed in our study is not solely attributable to biological sex but is largely influenced by gender roles, which are socially constructed and vary across individuals. In particular, the distinction between female, masculine, and undifferentiated roles suggests that gender identity and the internalization of societal expectations may significantly impact emotional resilience and psychological vulnerability. While women in general are often perceived as having a more pronounced connection to roles traditionally associated with caregiving and emotional expression, our findings indicate that those who adopt more masculine roles—typically associated with assertiveness and emotional stoicism—tend to experience higher levels of resilience and positive affect. Conversely, women who embrace undifferentiated or stereotypically feminine roles appear to face greater psychological challenges. This underscores the importance of considering not just sex but the broader social context of gender roles when assessing psychological well-being, particularly in academic settings where these roles may intersect with performance and personal development.

## Figures and Tables

**Figure 1 healthcare-13-00336-f001:**
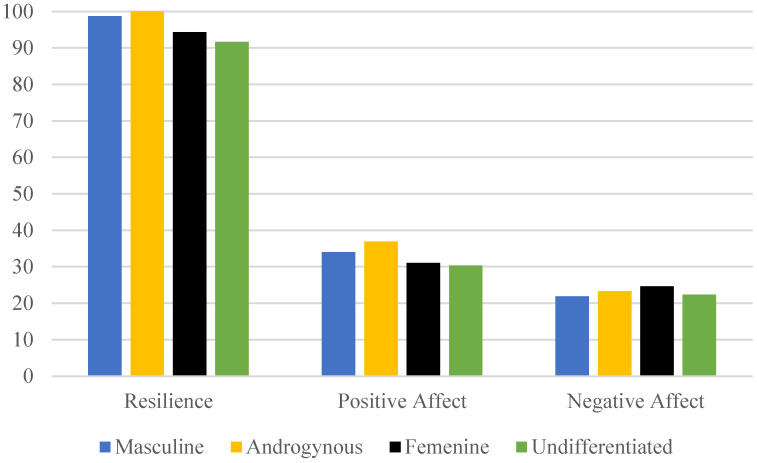
Levels of resilience, positive affect, and negative affect in female nursing students according to gender roles (n = 338).

**Figure 2 healthcare-13-00336-f002:**
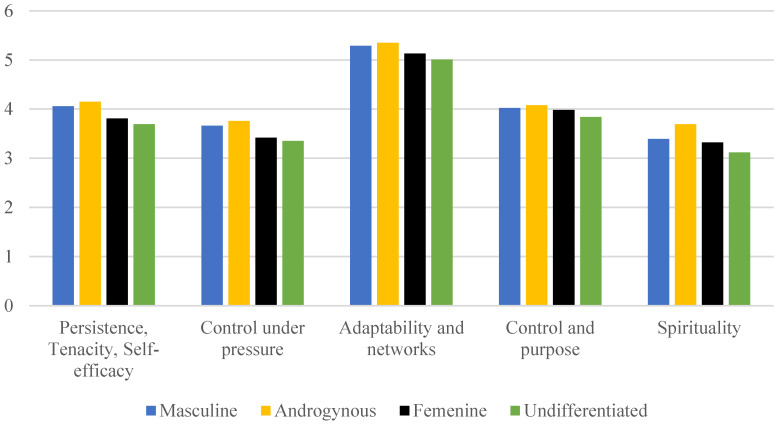
Results in subdimensions of resilience (CD-RISC) in female nursing students according to gender roles (n = 338).

**Figure 3 healthcare-13-00336-f003:**
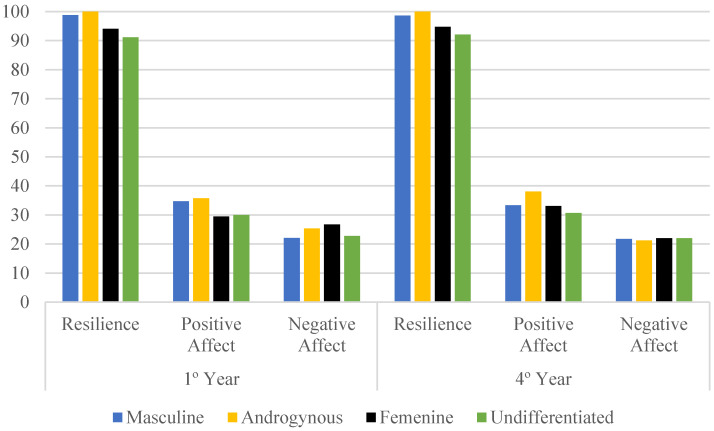
Differences between first-year and fourth-year nursing students in resilience, positive affect, and negative affect according to gender roles (n = 338).

**Table 1 healthcare-13-00336-t001:** Sociodemographic characteristics of the sample (n = 338).

		n	%	M	SD
Age				21.43	5.37
	<20	142	42.0		
	21–25	161	47.6		
	26–30	16	4.7		
	>30	19	5.6		
Course	1°	172	50.9		
	4°	166	49.1		
Marital status	Married	11	3.3		
	Single	274	81.1		
	Open relationship	53	15.7		
Living arrangements	With friends or fellow students	32	9.5		
	With parents or relatives	269	79.6		
	As a couple	19	5.6		
	Alone	15	4.4		
	None of the above	3	0.9		
Working status	No	220	65.1		
	Yes	118	34.9		
Occupation	High-level sport	48	14.2		
	Music or dance conservatory	9	2.7		
	Volunteering	58	17.2		
	Other university degree	16	4.7		
	Other studies	53	15.7		
	None of the above	154	45.6		
Resilience level (CD-RISC)				94.62	9.37
	Low	5	1.5		
	Medium	62	18.3		
	High	271	80.2		
Gender roles (BRSI)	Masculine	46	13.6		
	Feminine	80	23.7		
	Androgynous	48	14.2		
	Undifferentiated	164	48.5		

Note: CD-RISC: Connor–Davidson Resilience Scale; BSRI: Bem Sex-Role Inventory.

**Table 2 healthcare-13-00336-t002:** Sociodemographic characteristics according to Bem’s gender roles (n = 338).

		Masculine		Androgynous		Feminine		Undifferentiated		*p*-Value *
		n	%	n	%	n	%	n	%	
Age	<20	20	43.48	17	35.42	38	47.50	67	40.85	0.719
	21–25	21	45.65	22	45.83	35	43.75	83	50.61	
	26–30	2	4.35	4	8.33	4	5.00	6	3.66	
	>30	3	6.52	5	10.42	3	3.75	8	4.88	
Year course	1°	24	52.17	24	50.00	45	56.25	79	48.17	0.694
	4°	22	47.83	24	50.00	35	43.75	85	51.83	
Marital status	Married	1	2.17	4	8.33	3	3.75	3	1.83	0.357
	Single	39	84.78	38	79.17	66	82.50	131	79.88	
	Open relationship	6	13.04	6	12.50	11	13.75	30	18.29	
Living arrangements	With friends or fellow students	3	6.52	7	14.58	7	8.75	15	9.15	0.073
	With parents or relatives	39	84.78	32	66.67	63	78.75	135	82.32	
	As a couple	2	4.35	8	16.67	3	3.75	6	3.66	
	Alone	1	2.17	1	2.08	6	7.50	7	4.27	
	None of the above	1	2.17	0	0.00	1	1.25	1	0.61	
Working status	No	30	65.22	28	58.33	56	70.00	106	64.63	0.138
	Yes	16	34.78	20	41.67	24	30.00	58	35.37	
Occupation	High-level sport	8	17.39	8	16.67	12	15.00	20	12.20	0.609
	Music or dance conservatory	1	2.17	3	6.25	36	45.00	5	3.05	
	Volunteering	4	8.70	12	25.00	20	25.00	22	13.41	
	Other university degree	2	4.35	3	6.25	3	3.75	8	4.88	
	Other studies	11	23.91	8	16.67	9	11.25	25	15.24	
	None of the above	20	43.48	14	29.17	36	45.00	84	51.22	

* Note: The Chi-Square test was used in this analysis.

**Table 3 healthcare-13-00336-t003:** Differences in resilience, positive affect, and negative affect according to academic year and Bem’s gender roles (n = 338).

Year of Study	Study Variable	Masculine		Androgynous		Feminine		Undifferentiated		F	*p*-Value *
		M	SD	M	SD	M	SD	M	SD		
1° year	Resilience	98.79	7.46	101.54	7.57	94.09	9.99	91.18	9.02	10.55	<0.001
	Persistence, Tenacity, Self-Efficacy	4.10	0.41	4.17	0.39	3.74	0.63	3.65	0.57	8.34	<0.001
	Control under Pressure	3.67	0.45	3.74	0.37	3.43	0.42	3.32	0.42	8.66	<0.001
	Adaptability and Networks	5.27	0.35	5.35	0.50	5.15	0.66	5.00	0.55	3.36	0.02
	Control and Purpose	3.94	0.52	4.03	0.56	3.90	0.51	3.78	0.52	1.78	0.15
	Spirituality	3.35	0.70	3.92	0.96	3.47	1.13	3.28	0.93	2.76	0.04
	Positive affect	34.67	5.60	35.75	5.80	29.44	7.49	30.00	6.85	7.61	<0.001
	Negative affect	22.08	6.23	25.38	7.45	26.69	7.20	22.80	7.48	3.63	0.01
4° year	Resilience	98.59	5.94	101.08	7.99	94.71	7.07	92.06	9.74	8.65	<0.001
	Persistence, Tenacity, Self-Efficacy	4.02	0.40	4.13	0.43	3.90	0.40	3.73	0.52	5.82	<0.001
	Control under Pressure	3.64	0.38	3.79	0.41	3.40	0.40	3.38	0.45	7.18	<0.001
	Adaptability and Networks	5.32	0.44	5.35	0.54	5.09	0.47	5.01	0.60	3.48	0.02
	Control and Purpose	4.11	0.42	4.13	0.65	4.10	0.50	3.91	0.56	1.90	0.13
	Spirituality	3.43	1.09	3.46	1.11	3.13	1.01	2.97	1.03	2.05	0.11
	Positive affect	33.36	8.07	38.08	5.82	33.11	6.77	30.69	7.40	6.84	<0.001
	Negative affect	21.73	7.09	21.25	7.85	22.03	7.02	22.00	7.09	0.08	0.97

* Note: The ANOVA test was used in this analysis.

**Table 4 healthcare-13-00336-t004:** Correlations by year of nursing study (n = 338).

Year of Study	Study Variable	1	2	3	4	5
1° Year	1. Positive Affect	-				
	2. Negative Affect	−0.244 **	-			
	3. Masculine Roles	0.448 **	−0.077 **	-		
	4. Feminine Roles	0.057	0.292 **	0.084	-	
	5. Resilience	0.542 **	0.210 **	0.557 **	0.216 **	-
4° Year	1. Positive Affect	-				
	2. Negative Affect	−0.230 **	-			
	3. Masculine Roles	0.328 **	0.021	-		
	4. Feminine Roles	0.278 **	0.032	0.349 **	-	
	5. Resilience	0.398 **	−0.195 *	0.461 **	0.310 **	-

Note: * *p* < 0.05; ** *p* < 0.01.

## Data Availability

The datasets generated and/or analyzed during the current study are not publicly available due to data protection policy but are available from the corresponding author(s) upon reasonable request. This manuscript was drafted against the STROBE for observational research.

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
