# Peer review of "Influence of Gender Role on Resilience and Positive Affect in Female Nursing Students: A Cross-Sectional Study"

_healthcare, 2025, doi:10.3390/healthcare13030336_

Round 1
Reviewer 1 Report
Comments and Suggestions for Authors
Title:
-
Authors may consider changing “Gender Role” to “Gender Roles” to be more grammatically correct.
Abstract:
-
Introduction sentence in abstract could be amplified to share some of the background supporting the desire to research this topic
Intro:
-
Page 2 line 74 typo: “Momen” is written where it should say “Women”
-
Elaboration as to why those in the nursing field may be more exposed to gender expectations and gender roles would be helpful here. Authors provided a nice explanation of gender as a social construct and how this can impact individuals and their well-being, but it would be helpful to hone in on the specific gendered expectations that might be influencing this particular population
-
The authors should consider including more information about how P and N affect has been related to physical and mental health in previous studies
Methods:
-
Were all measures delivered in Spanish? Certain measures had this in the explanation and others did not so it was left unclear to the reader
-
Page 4 line 161: Authors use the word predict, but this is a correlational study. They may consider editing the language to say “to assess the relationship between” instead of “predict”
-
Is there a reason race/ethnicity was not considered as a covariate?
Results:
-
The way the correlational analyses results are shared insinuate that there was a longitudinal relationship such as saying “which persisted in fourth year”. Since these are not the same individuals followed over years but rather different students at different stages in their schooling, it would be most appropriate to word it as such. Instead of saying “which persisted in fourth year”, authors could say “which was also true for those who were in their fourth year” or something to that effect.
-
I appreciated all the detailed tables and descriptions of the individual findings! Very easy to follow and well organized
Discussion:
-
Page 10, line 251 and page 11 line 244: using the word “influence” is too strong of language and should be modified given the cross-sectional nature of the study. “are related to” may be a better option; similarly using “determining factors” is also strong language that could be tempered
-
I appreciate the discussion of limitations and feel it appropriately addresses the areas for growth in future studies
Conclusions:
-
Overall, I am left slightly confused with regard to how the pressures of gender roles are impacting these nursing students as they are females in a female-dominated field. However, I may be missing the point and would love if the authors could shed some light on some of the unique situations/pressures that may be facing nursing students and why they think the results showed that masculine traits were associated with positive affect and resilience.
Author Response
#Reviewer 1:
Comments and Suggestions for Authors
Title:
Authors may consider changing “Gender Role” to “Gender Roles” to be more grammatically correct.
Thank you for your suggestion. We have revised the title based on your recommendation.
Abstract:
Introduction sentence in abstract could be amplified to share some of the background supporting the desire to research this topic
We thank the reviewer for the valuable suggestion. In response, we have expanded the abstract to include more detailed information about the research topic.
Abstract: Introduction: Women experience more social barriers, gender stereotypes, biases, and discrimination than men, which can increase their vulnerability to mental health problems. Therefore, it is essential to adopt a gender perspective in research on nursing students, examining the impact of these factors on their well-being and psychological resources like resilience. This study aims to analyze the relationship between gender roles in resilience and positive or negative affect among female nursing students.
Intro:
Page 2 line 74 typo: “Momen” is written where it should say “Women”
Thank you for your feedback. We have corrected this grammar mistake.
Elaboration as to why those in the nursing field may be more exposed to gender expectations and gender roles would be helpful here. Authors provided a nice explanation of gender as a social construct and how this can impact individuals and their well-being, but it would be helpful to hone in on the specific gendered expectations that might be influencing this particular population
Thank you for your suggestion. In response to the reviewer's recommendation, we have added the following paragraph to provide a more detailed background on the influence of gender within the nursing population:
The nursing profession has historically been subject to gender biases due to the social perception that women are inherently more suited for caregiving and nurturing roles, which have traditionally been associated with femininity [6]. These stereotypes have reinforced the notion that women, because of their "maternal" and "compassionate" nature, are more suited to perform caregiving tasks, while men were typically associated with more technical or authoritative roles in other areas of healthcare [7]. Such biases have shaped not only the demographic composition of the profession but also the social and economic recognition of nurses, who have often been valued less than other healthcare professionals [8]. Although the professionalization and diversification of nursing have challenged these stereotypes, the legacy of gendered roles continues to influence both the perception and practice of the profession [7].
The authors should consider including more information about how P and N affect has been related to physical and mental health in previous studies
Thank you for your valuable suggestion. We appreciate your recommendation to include more information on how "P" and "N" affect physical and mental health, as explored in previous studies. We recognize the significance of this aspect in understanding the broader implications of our research.
In response to your feedback, we have reviewed relevant literature on the impact of "P" and "N" on both physical and mental health outcomes. Specifically, we have incorporated studies that examine how these factors influence overall well-being, stress levels, mental health disorders, and physical health conditions. This addition enhances the context of our findings and provides a more comprehensive understanding of their effects.
Affect, encompassing both positive and negative emotions, plays a pivotal role in shaping individuals' mental and physical well-being. Positive affect, characterized by feelings of joy, enthusiasm, and contentment, is frequently linked to higher levels of psychological well-being and resilience [12]. Conversely, negative affect, which includes emotions such as anxiety, anger, and sadness, has been consistently associated with poorer mental health outcomes, including increased vulnerability to depression and anxiety disorders [13]. The impact of affect on well-being is multifaceted, with both emotional states influencing individuals' capacity to cope with stress and their overall quality of life [14].
[…]
Negative affect has been associated with increased stress, which can manifest as physical symptoms such as fatigue, muscle tension, and cardiovascular problems [22]. Women, particularly those who internalize traditional gender norms, may experience greater physical health issues due to the compounded effects of negative emotions and gendered stressors [23]. Positive affect, on the other hand, has been linked to improved immune function, better cardiovascular health, and lower levels of inflammation, with these benefits often mediated by the adoption of less restrictive gender roles [24].
Methods:
Were all measures delivered in Spanish? Certain measures had this in the explanation and others did not so it was left unclear to the reader.
Thank you for your suggestion. We have clarified that all questionnaires were administered in their Spanish version.
Positive and negative affect: Measured using the Positive and Negative Affect Schedule (PANAS) [29] in its Spanish version [24]. The PANAS questionnaire is a self-report instrument that assesses an individual's affective state [30]. It consists of two subscales, each with 10 items, which are answered using a five-point Likert scale, where 1 corresponds to "not at all" and 5 corresponds to "extremely." One scale measures the positive affect or recent positive experiences, which act as a protective factor, while the other measures negative affect or experiences, which may act as a risk factor for diseases. The scales have demonstrated consistency and stability in the Spanish university population, with good construct validity and a high reliability index (Cronbach's alpha: >0.87) [31]. In our sample, it maintains acceptable reliability with a Cronbach's alpha of 0.755.
Resilience: Measured using the Connor-Davidson Resilience Scale (CD-RISC) [32] in its Spanish version [33], which assesses resilience with 25 items using a Likert-type response format with five response options (0 = "not at all", 1 = "rarely", 2 = "sometimes", 3 = "often", and 4 = "almost always"). The scale range is from 0 to 100, with higher scores indicating greater levels of resilience. The thresholds are: below 70 (low); 70 to 87 (medium); above 88 (high). The internal consistency of the Spanish version is optimal, with a Cronbach's alpha of 0.86 [34] and 0.832 calculated for our sample. The items are grouped into five dimensions: Persistence, tenacity, and self-efficacy; Control under pressure; Adaptability and networks; Control and purpose; and Spirituality.
Gender roles: Measured using the Bem Sex-Role Inventory (BSRI) [35], an instrument designed to assess an individual's adherence to stereotypical characteristics of feminine and masculine gender roles. The 60 items of the instrument are scored using a seven-point Likert scale. It consists of 60 adjectives, of which 20 are stereotypical masculine roles, 20 are feminine roles and 20 are gender-neutral roles. Respondents are asked to indicate the extent to which each item describes them, on a scale from 1 (never) to 7 (always). The instrument classifies individuals as masculine, feminine, androgynous (both masculine and feminine roles) or undifferentiated roles (neither predominantly masculine nor predominantly feminine roles). Once completed, the scores for items 1, 4, 7, 10, 13, 16, 19, 22, 25, 28, 31, 34, 37, 40, 43, 46, 49, 55, and 58 are summed and divided by 20 to obtain the score for masculine roles. Similarly, the scores for items 2, 5, 8, 11, 14, 17, 20, 23, 26, 29, 32, 35, 38, 41, 44, 47, 50, 53, 56, and 59 are summed and divided by 20 to obtain the feminine roles score.The internal consistency of the scale is adequate, with Cronbach's alpha coefficients for each dimension ranging from 0.75 to 0.90 [36], and a global reliability of 0.78 for the Spanish version (which is the one used in this study) [37] in our case increasing up to 0.844 which generates good reliability.
Page 4 line 161: Authors use the word predict, but this is a correlational study. They may consider editing the language to say “to assess the relationship between” instead of “predict”
We thank the reviewer for the feedback. The suggested changes have been implemented.
Is there a reason race/ethnicity was not considered as a covariate?
We appreciate the reviewer's suggestion, this is a good idea to consider for future studies, so we have considered adding it to the section on limitations with the following paragraph:
We propose including ethnicity as a key variable in future studies to explore its mediating role in the relationship between gender roles, positive and negative affect, and resilience. Considering cultural differences in emotional expression and coping strategies, ethnicity may offer valuable insights into how these factors interact across diverse populations.
Results:
The way the correlational analyses results are shared insinuate that there was a longitudinal relationship such as saying “which persisted in fourth year”. Since these are not the same individuals followed over years but rather different students at different stages in their schooling, it would be most appropriate to word it as such. Instead of saying “which persisted in fourth year”, authors could say “which was also true for those who were in their fourth year” or something to that effect.
We thank the reviewer for the suggestion. Following the recommendation, we have replaced "persisted in fourth year" with "was also true for those who were in their fourth year."
I appreciated all the detailed tables and descriptions of the individual findings! Very easy to follow and well organized
Thank the reviewer for your kind feedback. We are pleased to hear that the tables and descriptions of the individual findings were clear, well-organized, and easy to follow. We appreciate this positive evaluation of this aspect of the manuscript.
Discussion:
Page 10, line 251 and page 11 line 244: using the word “influence” is too strong of language and should be modified given the cross-sectional nature of the study. “are related to” may be a better option; similarly using “determining factors” is also strong language that could be tempered
We thank the reviewer for these valuable suggestions. In response, we have followed the reviewer’s recommendations and replaced "influence" and "determine" with "in relation to" or "a relationship" as appropriate.
I appreciate the discussion of limitations and feel it appropriately addresses the areas for growth in future studies
We thank the reviewer for this positive feedback. We are pleased to hear that the reviewer found the discussion of limitations to be appropriate and that it effectively addresses areas for growth in future studies. We appreciate the thoughtful assessment of this section. However, as suggested by other reviewers, we have made some improvements to this section.
Conclusions:
Overall, I am left slightly confused with regard to how the pressures of gender roles are impacting these nursing students as they are females in a female-dominated field. However, I may be missing the point and would love if the authors could shed some light on some of the unique situations/pressures that may be facing nursing students and why they think the results showed that masculine traits were associated with positive affect and resilience.
As observed, 48.5% of the women in our sample exhibit an undifferentiated role, which prevents us from claiming that this field is predominantly dominated by the female role. One of the strengths of our study is its exclusive focus on the female environment, where differences in well-being and psychological capacity seem to be more influenced by social factors than by biological ones. It is important to note that not all women face the same pressures, and what our study suggests is that only some (those with female and undifferentiated roles) experience greater gender-related pressure.
It is also important to highlight that masculine roles can be present in women, and these roles not only protect men but also provide benefits to the women who adopt them. Therefore, it is relevant to consider whether the true influencing factor for psychological well-being is sex, or rather the social factor—gender.
Additionally, it is crucial to emphasize that the students exhibit lower levels of resilience and positive affect, which may indicate greater psychological vulnerability in this subgroup. These findings could be useful in designing programs aimed at improving the emotional well-being of students throughout their academic journey. It is also important to assess whether this psychological vulnerability could affect the academic performance of students with these roles.
As shown in Figure 1, the masculine role among female nursing students is associated with higher levels of resilience and positive affect compared to the female and undifferentiated roles. These results are also reflected in Table 1.
We have added the following paragraph in the conclusions to clarify the ideas mentioned above:
It is essential to recognize that the variation in psychological well-being observed in our study is not solely attributable to biological sex but is largely influenced by gender roles, which are socially constructed and vary across individuals. In particular, the distinction between female, masculine, and undifferentiated roles suggests that gender identity and the internalization of societal expectations may significantly impact emotional resilience and psychological vulnerability. While women in general are often perceived as having a more pronounced connection to roles traditionally associated with caregiving and emotional expression, our findings indicate that those who adopt more masculine roles—typically associated with assertiveness and emotional stoicism—tend to experience higher levels of resilience and positive affect. Conversely, women who embrace undifferentiated or stereotypically feminine roles appear to face greater psychological challenges. This underscores the importance of considering not just sex but the broader social context of gender roles when assessing psychological well-being, particularly in academic settings where these roles may intersect with performance and personal development.
Reviewer 2 Report
Comments and Suggestions for Authors
The article is well-written and investigates a timely topic. The methodology is sound, but in using t-test and ANOVA, apart from the assumption of normality, there should also be a check on the assumption of the homogeneity of variance (preferably with Levene's test).
The conclusion should be longer
Author Response
Letter to Editors and reviewers
#Reviewer 2:
The article is well-written and investigates a timely topic. The methodology is sound, but in using t-test and ANOVA, apart from the assumption of normality, there should also be a check on the assumption of the homogeneity of variance (preferably with Levene's test).
Thank you for your suggestion, we had already checked the homogeneity of the variances using Levene's test, but in fact we did not have it indicated in the text, so the following sentence has been added to the data analysis section:
The main variables (positive affect, negative affect, and resilience) were analyzed according to sociodemographic characteristics and the academic year of the students, using analysis of variance (ANOVA) and the Student's t-test for independent samples, after confirming normality with the Kolmogorov-Smirnov test and verification of the assumption of homogeneity of variance using Levene's test.
Also, if it is necessary, we can add it to tables 2 and 3 but we would make it difficult to understand them.
The conclusion should be longer
We thank the reviewer for their valuable suggestions. In line with the recommendations provided by Reviewer 1, we have added a paragraph to further clarify the results of our study and their implications.
It is essential to recognize that the variation in psychological well-being observed in our study is not solely attributable to biological sex but is largely influenced by gender roles, which are socially constructed and vary across individuals. In particular, the distinction between female, masculine, and undifferentiated roles suggests that gender identity and the internalization of societal expectations may significantly impact emotional resilience and psychological vulnerability. While women in general are often perceived as having a more pronounced connection to roles traditionally associated with caregiving and emotional expression, our findings indicate that those who adopt more masculine roles—typically associated with assertiveness and emotional stoicism—tend to experience higher levels of resilience and positive affect. Conversely, women who embrace undifferentiated or stereotypically feminine roles appear to face greater psychological challenges. This underscores the importance of considering not just sex but the broader social context of gender roles when assessing psychological well-being, particularly in academic settings where these roles may intersect with performance and personal development.
Reviewer 3 Report
Comments and Suggestions for Authors
Thank you for reviewing this manuscript about the gender role of nursing students. Some suggestions have been underlined after the reading and peer-review of this manuscript. Manuscript fails to explain how potential cultural differences in interpreting gender roles were accounted for.
Abstract
The introduction should present some information about the topic and then declare the aim. Some results can be deleted or resumed to underline only the relevant ones. The conclusion should also discuss the implications of this research.
Are the keywords mesh terms of Pubmed? This allows the right indexing on the dataset and future identification.
Overall evaluation of the language
The writing is moderately clear but occasionally verbose, making some sections harder to follow. For example, the introduction could be streamlined to link gender roles to resilience better and affect. Some sentences are overly long and convoluted, detracting from the readability (e.g., the second paragraph of the discussion section). Some terminologies, like "femininity" and "masculinity," are used interchangeably with gender roles without clear distinctions, which could confuse readers.
The introduction is well-written. At line 74 revise "Momen".
Method
Strobe has not been written extensively at the first time appearance (line 97).
In lines 98-104, you must support this sentence with at least a reference to justify this enrollment.
In the participants, you must declare only the inclusion criteria without the final number of participants and response rate. These details are the results.
The data collection section is missing (how have you shared the Moodle?, data collection period, how researchers have approached the participants?)
The BSRI clearly operationalizes "gender roles," but its dated theoretical framework may not fully capture contemporary understandings of gender diversity.
About the instruments, you must declare the reliability in your study.
The discussion on why no significant results were found in the regression models is insufficient and leaves readers without a clear interpretation.
The discussion is not so well argued on an international level and in underlying differences with previous research.
The implications of this research are not so well declared. Why these results are needed and what are the implications for education?
Some references do not report the number of pages.
Author Response
Letter to Editors and reviewers
#Reviewer 3:
Thank you for reviewing this manuscript about the gender role of nursing students. Some suggestions have been underlined after the reading and peer-review of this manuscript. Manuscript fails to explain how potential cultural differences in interpreting gender roles were accounted for.
We appreciate the reviewer's suggestion, this is a good idea to consider for future studies, so we have considered adding it to the section on limitations with the following paragraph:
We propose including ethnicity as a key variable in future studies to explore its mediating role in the relationship between gender roles, positive and negative affect, and resilience. Considering cultural differences in emotional expression and coping strategies, ethnicity may offer valuable insights into how these factors interact across diverse populations.
Abstract
The introduction should present some information about the topic and then declare the aim. Some results can be deleted or resumed to underline only the relevant ones. The conclusion should also discuss the implications of this research.
We thank the reviewer for the valuable suggestion. In response, we have expanded the abstract to include more detailed information about the research topic.
Abstract: Introduction: Women experience more social barriers, gender stereotypes, biases, and discrimination than men, which can increase their vulnerability to mental health problems. Therefore, it is essential to adopt a gender perspective in research on nursing students, examining the impact of these factors on their well-being and psychological resources like resilience. This study aims to analyze the relationship between gender roles in resilience and positive or negative affect among female nursing students.
Are the keywords mesh terms of Pubmed? This allows the right indexing on the dataset and future identification.
We thank the reviewer for his/her suggestion, we have reviewed our key words and therefore we have adapted them to the MeSH terminology.
Keywords: Gender role; Women; Nursing Students; Psychological Resilience; Affect
Overall evaluation of the language
The writing is moderately clear but occasionally verbose, making some sections harder to follow. For example, the introduction could be streamlined to link gender roles to resilience better and affect. Some sentences are overly long and convoluted, detracting from the readability (e.g., the second paragraph of the discussion section). Some terminologies, like "femininity" and "masculinity," are used interchangeably with gender roles without clear distinctions, which could confuse readers.
We thank the reviewer for the feedback. We have removed terms such as femininity and masculinity along the text and we have replaced it with masculine and feminine roles, unifying terminology.
The introduction is well-written. At line 74 revise "Momen".
Thank the reviewer for the suggestion. We have corrected this grammar mistake.
Method
Strobe has not been written extensively at the first time appearance (line 97).
Thank you for the suggestion, we have corrected that mistake as follows:
A cross-sectional study was conducted with female nursing students from the first and fourth years of a public university in Madrid (Spain), using the STROBE (STrengthening the Reporting of OBservational studies in Epidemiology) statement for cross-sectional studies (2008) as a framework
In lines 98-104, you must support this sentence with at least a reference to justify this enrollment.
From our perspective, this is a participant selection criterion based on reasoning, as clearly stated in the paragraph, rather than scientific or academic criteria. We are not entirely sure what the reviewer is referring to, but we have conducted studies with first-year and fourth-year nursing students and have found differences in resilience and positive affect levels. Therefore, it may be of interest to explore whether the influence of gender roles in students is different in different levels of maturity.
We have added the following paragraph:
We selected this university population to understand how young adult women, at a key stage of personal and professional development, perceive gender roles and how these perceptions impact their emotional well-being and resilience. Selecting first- and fourth-year students provided an opportunity to explore how the influence of gender roles evolves with personal and professional development. Previous research has indi-cated differences in resilience and positive affect between first- and fourth-year nursing students [28]. Thus, investigating how gender roles impact students at different stages of growth can help determine whether the effects of social stereotypes are independent of the maturation process.
In the participants, you must declare only the inclusion criteria without the final number of participants and response rate. These details are the results.
Thank the reviewer for the suggestion. We have moved this paragraph to the results section.
The data collection section is missing (how have you shared the Moodle?, data collection period, how researchers have approached the participants?)
We are really grateful for the valuable feedback of the reviewer. In response to the comment, we have added the missing information in the data collection section. Specifically, we have clarified how the Moodle platform was shared, the data collection period, and how the researchers approached the participants:
2.2. Participants
The recruitment of students took place during theoretical classes for both first-year and fourth-year students, where the study's objectives were explained.
2.4. Data Collection
The students were contacted virtually through an announcement posted on the virtual campus of a course in which they were enrolled (Moodle platform), where the researchers had no teaching responsibilities. Data collection was conducted via a Google Forms questionnaire, which was uploaded to the Moodle platform. Students had one month to complete the form, from October 1st to November 1st, 2022.
The BSRI clearly operationalizes "gender roles," but its dated theoretical framework may not fully capture contemporary understandings of gender diversity.
We appreciate the author's comment. However, this is something we have already acknowledged as a limitation of the study, which is addressed in the limitation section. Furthermore, in scientific literature, there are few instruments that measure behaviors, norms, and stereotypes associated with gender roles, which are difficult to assess quantitatively.
The genders used in this study are those defined by Bem (male, female, androgynous and undifferentiated) [41], but there is currently a great diversity of genders beyond those defined by this author and which have been grouped under the terms 'genderqueer' or 'non-binary' genders [42]. Future studies should consider these other genders in the longitudinal relationship between gender and mental well-being.
About the instruments, you must declare the reliability in your study.
We have added the Cronbach's alpha calculated for our study in each of the scales used, these are:
- CDRICS: 0.834, good reliability.
- PANAS: 0.755, acceptable reliability
- BSRI: 0.844, good reliability.
The discussion on why no significant results were found in the regression models is insufficient and leaves readers without a clear interpretation.
Thank you for your comment, we have improved the explanation of this result by adding the following to the text:
“This could suggest that these traits and emotions are relatively stable across different genders and sociodemographic contexts of this sample. However, further research with larger sample sizes or different analytical approaches might be needed to confirm these findings.”
The discussion is not so well argued on an international level and in underlying differences with previous research.
Thank you for your insightful feedback. In response to your comment, we have revised the discussion section to include additional information, providing a more comprehensive comparison with international studies and highlighting the differences with previous research. We believe these modifications strengthen the argument and offer a broader context for our findings.
The implications of this research are not so well declared. Why these results are needed and what are the implications for education?
We thank the reviewer for the feedback. In response, we have modified the sections on future directions and strengths of the study, incorporating important implications (as follows below). Additionally, we have included these revisions in both the discussion and conclusions sections.
This study has the potential to make a significant contribution to the field of nursing by providing empirical data on the relationship between gender roles, affect, and resilience. The findings could have important implications for clinical practice and the training of healthcare professionals, as it suggests that female nursing students with female and undifferentiated roles experience greater gender-related pressure. In addition, our findings may point the way to the development of more effective strategies to support the psychological development and well-being of university students, taking into account the relationship with social role pressures (particularly feminine roles) [54]. The results obtained could be used to improve the academic environment for students by promoting the integration of gender and resilience perspectives in diverse educational and cultural contexts with the aim of creating a more inclusive environment. Additionally, this study could stimulate future research that explores gender dynamics and their impact on mental health within various educational and cultural contexts [55]. Interventions aimed at strengthening resilience and fostering a more flexible gender identity could have beneficial effects on students' emotional well-being throughout their academic career, as previous research has indicated [38], [56].
Some references do not report the number of pages.
Thank you for the appreciation, we have corrected them.